

# Contact zone of slow worms *Anguis fragilis* Linnaeus, 1758 and *Anguis colchica* (Nordmann, 1840) in Poland

Grzegorz Skórzewski[1], Bartosz Borczyk[2], Stanisław Bury[3], Daniel Kulik[1,4] and Jan Kotusz[1]

[1] Museum of Natural History, University of Wroclaw, Wroclaw, Poland
[2] Department of Evolutionary Biology and Conservation of Vertebrates, University of Wroclaw, Wroclaw, Poland
[3] Department of Comparative Anatomy, Jagiellonian University Cracow, Kraków, Poland
[4] Laboratory of Non-Mendelian Evolution, Institute of Animal Physiology and Genetics CAS, Liběchov, Czech Republic

Corresponding author
Grzegorz Skórzewski,
grzegorz.skorzewski@uwr.edu.pl

## ABSTRACT

**Background**. Legless lizards, the slow worms of the genus *Anguis*, are forming secondary contact zones within their Europe-wide distribution.

**Methods**. We examined 35 populations of *A. fragilis* and *A. colchica* to identify the level of morphological and genetic divergence in Poland. We applied a conventional study approach using metric, meristic, and categorial (coloration) features for a phenotype analysis, and two standard molecular markers, a mitochondrial (NADH-ubiquinone oxidoreductase chain 2; *ND2*) and a nuclear (V(D)J recombination-activating protein 1; *RAG1*) one.

**Results**. We found clear differences between *A. fragilis* and *A. colchica* in molecular markers and phenotype—in meristic features, *e.g.,* ear opening, number of scales rows around the body, and higher than so far known diversity in *ND2* and *RAG1* haplotypes. The presence of five hybrids was detected in three populations in the Polish part of the European contact zone. In all hybrids, homozygous alleles of *RAG1* were detected, which suggests a back-crossing within the genus.

**Conclusions**. The ability to produce fertile offspring by *A. fragilis* x *A. colchica* hybrids shows inefficient mechanisms of reproductive isolation of the two legless lizards. The hybrids were indistinguishable from parental species in head proportions (principal components and discriminant analyses) but more resembling *A. colchica* in meristic traits.

## INTRODUCTION

In animals, hybrid zones are usually considered abrupt discontinuities between differentiated groups of populations that are relatively homogenous over large areas (*Harrison, 1993*). Such discontinuities were found in the distribution of well-known legless lizards in Europe, slow worms of the genus *Anguis*. Although European reptiles have been intensely investigated for years, the major issues concerning the slow worms' contact zones, such as their origin, shape, dynamics of hybridization or fate are not fully

understood. At the beginning of the twenty-first century, using molecular methods at least five slow worm species were identified within the range that was long believed to be occupied by a single species, *A. fragilis* (*Gvoždík et al., 2010*; *Gvoždík et al., 2013*). Moreover, hybridisation in this genus was revealed between the species pairs: *A. fragilis*—*A. veronensis* in Nothern Italy and SouthernFrance (*Gvoždík et al., 2013*; *Dufresnes et al., 2023*), *A. fragilis*—*A. colchica* in Hungary (*Szabó & Vörös, 2014*), Czechia and Slovakia (*Gvoždík et al., 2015*; *Šifrová, 2017*; *Benkovský et al., 2021*; *Harca, 2021*). Another region of slow worms co-occurrence and potential gene flow was reported from Montenegro between *A. graeca* and *A. fragilis* (*Mikulíček et al., 2018*). Similar species pair was described in Greece (*A. cephallonica* and *A. graeca*) but no hybridisation was recorded there (*Koppitz, 2018*; *Thanou et al., 2021*). Although the diversification of those taxa was estimated to originate in the late Miocene (*Gvoždík et al., 2023*), the pattern of their current distribution was shaped mainly by the last Pleistocene glaciation and subsequent global warming causing habitat transformations and dispersion of the mentioned reptiles from refuges like many organisms in the northern hemisphere (*Hewitt, 2000*). The contemporary variability in the mitochondrial DNA of slow worms was mostly shaped by topography of the Balkans (*Jablonski et al., 2016*). The same paper highlights the crucial role of refugia-within-refugia model in the post-glacial recolonization of vast area of Europe.

The distribution area of the distinguished *Anguis* taxa has recently been elaborated by *Jablonski et al. (2021)*. A significant part of this area was defined as a grey zone—where taxonomically unassined records prevail. The grey zone, which runs from the Sea of Marmara to the Gulf of Finland in the Baltic Sea includes some secondary contact zones between slow worm species. This picture develops the former morphology-based concept of *Voipio (1962)*, who observed abrupt changes in the frequency of some meristic character states (*i.e.,* number of scales rows around the central part of body, presence/absence of ear opening, types of prefrontal shields contact) along a line extending from the Carpathian Mountains northeast to the Baltic coast. In Poland, two slow worm species are known from this area, *A. fragilis* in the western and *A. colchica* in the eastern part of the country (Figs. 1–3) (*Gvoždík et al., 2010*; *Skórzewski, 2017*; *Jablonski et al., 2017*). The entire distribution of *A. fragilis* comprises a vast area of Western Europe from the Iberian Peninsula and British Isles to Central and south-eastern Europe, while *A. colchica* occurs from Central Europe to Russia as far as behind the Urals, northern Turkey, Caucasus and northern Iran (*Petzold, 1971*; *Dely, 1974*; *Völkl & Alfermann, 2007*; *Gvoždík et al., 2010*). The intra-generic hybridization in the Polish contact zone was suggested a while ago (*Jablonski et al., 2017*), but the hybrid specimens have not yet been reported.

The taxonomic dispute in Poland was closed for decades following the opinion of *Juszczyk (1987)* of solely *A. fragilis* distributed throughout the country. Intriguingly, the Polish part of the grey zone is a continuation of the Czech-Slovak hybrid zone, in which hybridization has been genetically confirmed (*Benkovský et al., 2021*). In the mentioned study, a link between phenotypic diversity and genetic variability of the two species and their hybrids was investigated. For the first time, the morphological comparison of several *A. colchica* and *A. fragilis* populations was preceded by the genetic identification of considerable number of specimens, especially from the potential hybrid zone. It allowed the

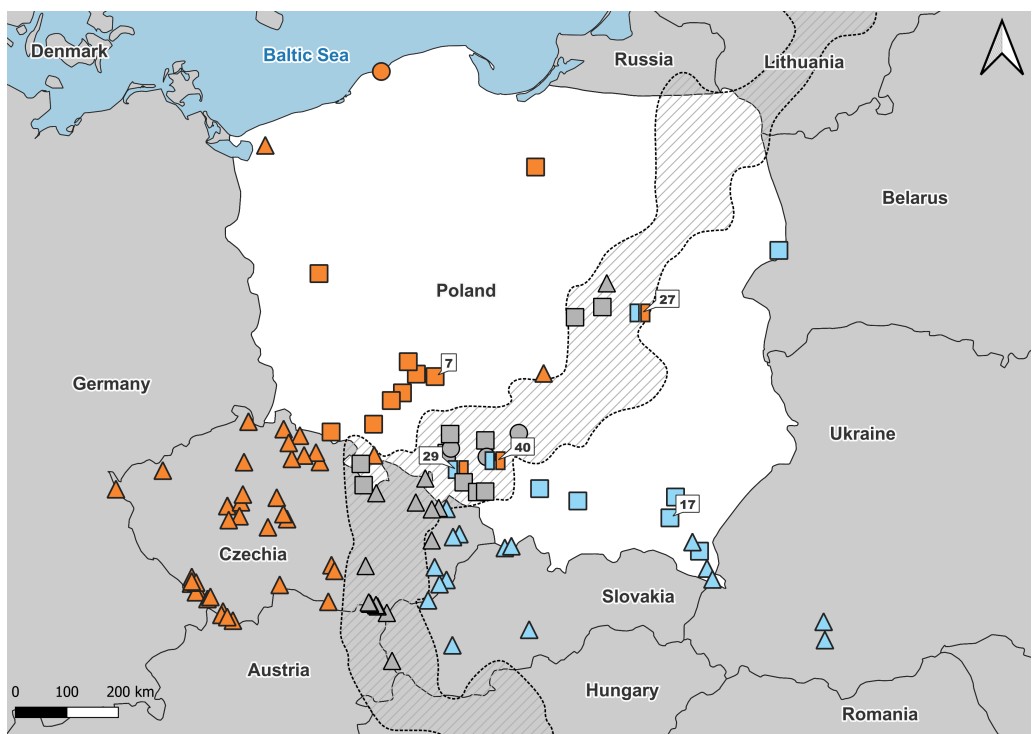

**Figure 1 Samples used in the study.** Number refers to population ID in Table 1. Squares, specimens used in genetic and morphological analyses; circles, specimens used in genetic analyses only; triangles, specimens used in morphological analyses only. Blue, *A. colchica*; orange, *A. fragilis*; blue-orange map-markers indicate hybrid specimens; grey, grey zone specimens. Dashed area "grey zone" (*Jablonski et al., 2021*). Samples from southern Europe and Asia are not shown.

authors to outline a reliable evolutionary scenario of differentiation in some morphological characters, concluding that *A. fragilis* bears more plesiomorphic traits, and hybrids resemble *A. fragilis* more. It seems especially important as some morphological traits, taken as species-diagnostic like blue spots in the dorsal part of the body, differ between the two species by frequency of occurrence (*Bury et al., 2023*).

In this study, first we searched for hybrid specimens in the putative Polish grey zone, which is a continuum of the Czech-Slovak zone, by analysing the diversity of two standard molecular markers, a mitochondrial (*ND2*) and a nuclear (*RAG1*) one. Second, we provide phenotypic descriptions of *A. colchica* and *A. fragilis*, supplementing the morphological diagnoses currently used. We focused on metric, meristic and categorial (colouration) features for phenotype analysis. Third, we searched for morphological variability of specimens originating from the contact zone, especially focusing on molecularly-identified hybrids.

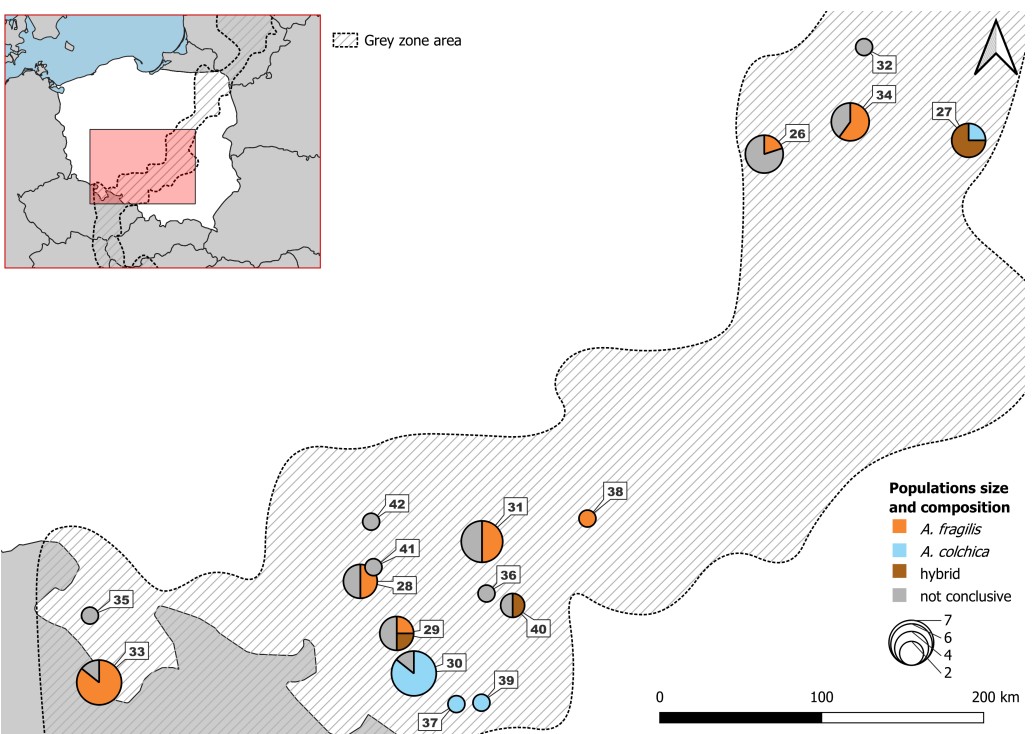

**Figure 2  Samples from Polish part of 'grey zone' used in the study.** Detailed usage of each individuals in Table S5B.

## MATERIALS & METHODS

### Sampling and species identification

The materials used in the study ($n = 251$) originated from two sources: field-collected individuals from April 2015 to August 2017, representing 35 populations of Poland, and specimens from museum collections (79 populations), representing mostly neighbouring areas in Europe (Table 1; Fig. 1). The museum specimens were taken for morphological examination only. Species affinity of collected material was set in a three-way approach. Individuals from the field were identified genetically using two molecular markers: (1) mitochondrial DNA gene NADH dehydrogenase subunit 2, (*ND2*) ($n = 89$) (*Gvoždík et al., 2013*) and (2) nuclear DNA gene *RAG1* ($n = 71$) (*Szabó & Vörös, 2014*). At first, the individuals were classified using the BLAST tool on the *ND2* sequence and then verified according to the results of phylogenetic analyses. The remaining part of the field material was identified as *A. fragilis vs. A. colchica* based on distributional criterion (*Jablonski et al., 2021*). Individuals from grey zone of incomplete molecular identification (by less than two markers) were assigned to GZ group and as such subjected to morphological analyses (Table S5A, Fig. 2). Moreover, we checked the species identification using standard morphological characters for European slow worms, including scale rows in the central part of the body and ear opening presence (Table S1A). The sexes of field-collected individuals were determined

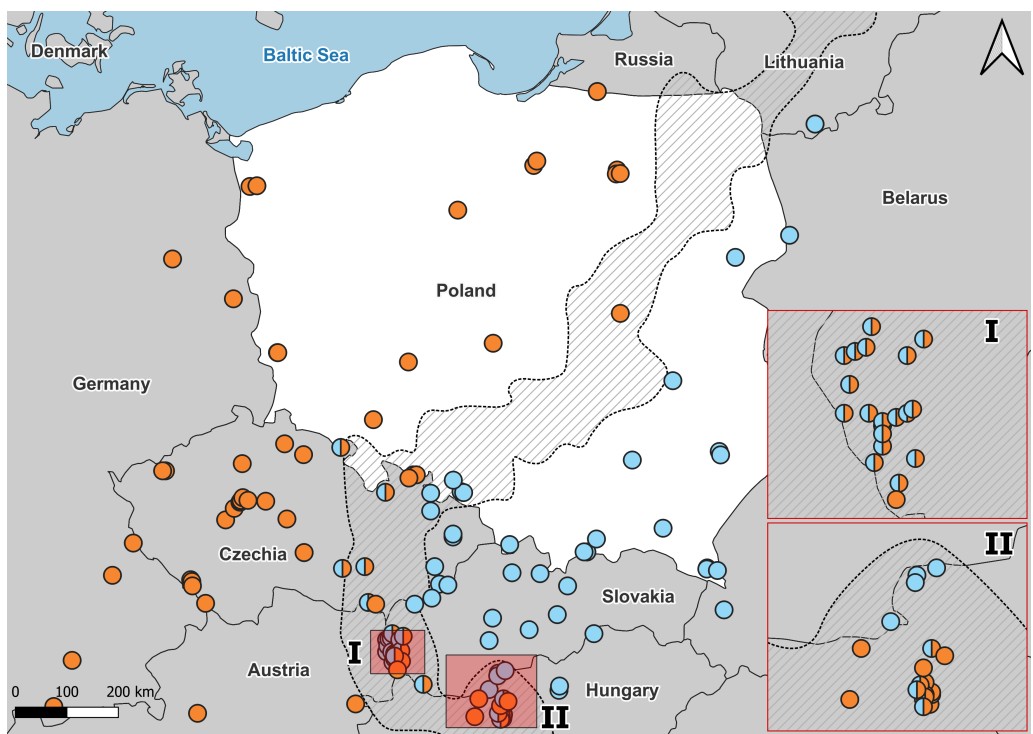

**Figure 3** **Distribution of *A. colchica*, *A. fragilis* and detected hybrid zone in Central Europe based on literature data.** Blue, *A. colchica*; Orange, *A. fragilis*. Combined molecular data from *Gvoždík et al. (2010)*, *Gvoždík et al. (2013)*, *Gvoždík et al. (2021)*, *Szabó & Vörös (2014)*, *Jablonski et al. (2017)*, and *Benkovský et al. (2021)*. I, hybrid zone in Hungary; II, Hybrid zone in Slovakia.

by dissections, and museum specimens were determined based on coloration pattern and other sexually dimorphic traits (*Juszczyk, 1987*; *Sos & Herczeg, 2009*).

Individuals were collected in the field (Poland) thanks to the permission of the General Director of Environment Protection in Poland (No. DZP WG. 6431.02.4.2015.JRP). The number of 78 individuals were euthanized according to the protocol of *Conroy et al. (2009)*; no *in vivo* experiments were performed. In the first step of euthanasia, 1% MS222 solution was injected into the specimen's coelom. In the second step, after the body lost the righting reflex completely, the 50% MS222 solution was injected. The injection volume was adjusted to the specimen weight. Next, secondary physical euthanasia by removing internal organs was performed (obtained organs were used for dietary and parasite infection survey). Details on euthanasia protocol, including solutions preparation, timing, and outcomes are in *Conroy et al. (2009)*. The used specimens were deposited in the Museum of Natural History of the University of Wrocław.

## Molecular laboratory procedures and phylogenetic analyses
Total genomic DNA was extracted using GeneMATRIX TISSUE extraction kits following the manufacturer's protocols (EURX). Two protein-coding gene fragments were amplified: an 732 bp fragment of the mitochondrial NADH dehydrogenase 2 gene (*ND2*), and a 1010

**Table 1 Materials used in this study.**

| Country | Species | Population ID | Locality | Coordinates | | n | Museum voucher ID | ND2/RAG1 code | Genotype ND2/RAG1 | ND2 haplogroups | GenBank accession number ND2/RAG1 |
|---|---|---|---|---|---|---|---|---|---|---|---|
| | | | | N | E | | | | | | |
| Poland | A. fragilis | | | | | | | | | | |
| | | 1 | Bardo, | 50.503 | 16.744 | 1 | MNHW-Reptilia-0280 | – | – | – | |
| | | 2 | Borowa Oleśnicka | 51 .190 | 17.280 | 1 | MNHW-Reptilia-0330 | – | – | – | |
| | | 3 | Byków | 51.191 | 17.237 | 4 | | By2016 | F/F (2) | ICE | PP549462–PP549463/PP525188–PP525189 |
| | | 4 | Goszcz | 51.396 | 17.480 | 1 | | GO2016 | – | ICE | PP549472 |
| | | 5 | Iława | 53.597 | 19.561 | 3 | | I2017_1-3 | F/F (2); F/ -(1) | ICE | PP549473–PP549475/PP525194–PP525195 |
| | | 6 | Janiszów | 50.761 | 15.991 | 1 | | J2016_1 | F/F (1) | ICE | PP549476/PP525196 |
| | | 7 | Ligota | 51.369 | 17.807 | 6 | | KG2015 | F/F (6) | ICE | PP549478–PP549483/PP525200–PP525205 |
| | | 8 | Piotrków Trybunalski | 51.404 | 19.702 | 3 | | – | – | – | |
| | | 9 | Pszczew | 52.478 | 15.780 | 7 | MNHW-Reptilia-0316,17,18 | PS2014/2017 | F/F (5); F/- (1); | ICE | PP549503–PP549508/PP525218–PP525222 |
| | | 10 | Ruda Milicka | 51.531 | 17.338 | 7 | MNHW-Reptilia-0333,34,35 | DB2014/2015 | F/F (2); F/- (1) | ICE | PP549465–PP549467/PP525191 |
| | | 11 | Sulistrowiczki | 50.847 | 16.733 | 8 | MNHW-Reptilia-0310 | S2015 | F/F (4); -/F (1) | ICE | PP549509–PP549512/PP525223–PP525227 |
| | | 12 | Ustka | 54.575 | 16.868 | 1 | | U2016 | F/F (1) | ICE | PP549516/PP525228 |
| | | 13 | Wrocław | 51.107 | 17.042 | 8 | MNHW-Reptilia-0248 | W2015/2016 | F/F (4) | ICE | PP549517–PP549520/PP525229–PP525231 |
| | | 14 | Wysoka Kamieńska | 53.821 | 14.843 | 1 | | – | – | – | |
| | A. colchica incerta | | | | | | | – | – | – | |
| | | 15 | Kłaj | 49.993 | 20.299 | 3 | | Kl2015 | C/C (1) | I | PP549438/PP525167 |
| | | 16 | Krzeszowice | 50.131 | 19.632 | 1 | | KY2015 | C/C (1) | I | PP549439/PP525168 |
| | | 17 | Lutcza | 49.803 | 21.907 | 8 | | Lu2015/2017 | C/C (8) | I; IV | PP549447–PP549454/PP525174–PP525181 |
| | | 18 | Myszkowce - Bóbrka | 49.426 | 22.423 | 3 | | M2016 | C/C (1) | I | PP549455/PP525182 |
| | | 19 | Pogorzelce | 52.724 | 23.809 | 1 | | BL2016 | C/C (1) | III | PP549432/PP525161 |
| | | 20 | Przełęcz Rycerska | 49.463 | 19.024 | 2 | | – | – | – | |
| | | 21 | Rzeszów-Czudec | 50.037 | 22.004 | 3 | | RZ2017 | C/C (1) | IV | PP549457/PP525183 |
| | | 22 | Sękowiec | 49.229 | 22.554 | 7 | MNHW-Reptilia-247 | – | – | – | |
| | | 23 | Ujsoły | 49.483 | 19.139 | 2 | | – | – | – | |
| | | 24 | Ustrzyki Górne | 49.105 | 22.650 | 2 | AC 73505 (MIZ) | – | – | – | |
| | | 25 | Załuż | 49.530 | 22.300 | 1 | AC 73506 (MIZ) | – | – | – | |
| | Grey zone | | | | | | | – | – | – | |
| | | 26 | Bartniki | 52.011 | 20.250 | 5 | | Ba2015 | F/F (1); F/- (1) | ICE | PP549461–PP549462/PP525187 |
| | | **27** | **Celestynów** | **52.058** | **21.384** | **4** | | **C2017** | **C/C (1); C/? (3)** | **III** | **PP549433–PP549436/PP525162–PP525165** |
| | | 28 | Górażdże | 50.528 | 18.009 | 4 | | G2015/2016 | F/F (2); F/- (2) | ICE | PP549468–PP549471/PP525193 |
| | | **29** | **Kędzierzyn Koźle** | **50.344** | **18.211** | **4** | | **KK2016/2017** | **F/F (1); C/F (1); -/F (2)** | **ICE; I** | **PP549437;477/PP525166; 197-199** |

**Table 1** (*continued*)

| Country | Species | Population ID | Locality | Coordinates | | n | Museum voucher ID | *ND2/RAG1* code | Genotype | *ND2* haplogroups | GenBank accession number *ND2/RAG1* |
|---|---|---|---|---|---|---|---|---|---|---|---|
| | | | | N | E | | | | *ND2/RAG1* | | |
| | | 30 | Kuźnia Raciborska | 50.202 | 18.306 | 7 | | KR2017 | C/C (6); C/- (1) | I | PP549440–PP549446 /PP525169–PP525173 |
| | | 31 | Lubliniec | 50.668 | 18.684 | 6 | | Lb2016 | F/F (3); F/- (3) | ICE | PP549484–PP549489 /PP525206–PP525208 |
| | | 32 | Łomna | 52.375 | 20.804 | 1 | AF 73628 (MIZ) | – | – | – | |
| | | 33 | Niemojów | 50.170 | 16.560 | 7 | | N2015 | F/F (6); F/- (1) | ICE | PP549490–PP549496 /PP525209–PP525214 |
| | | 34 | Podkowa Leśna | 52.121 | 20.727 | 5 | AF73520 (MIZ) | PL2015/2016 | F/F (3), F/- (1) | ICE | PP549498–PP549501 /PP525215–PP525217 |
| | | 35 | Polanica Zdrój | 50.407 | 16.509 | 1 | | PZ2017 | F/- (1) | ICE | PP549502 |
| | | 36 | Połomia | 50.485 | 18.709 | 1 | | OL2016 | F/- (1) | ICE | PP549497 |
| | | 37 | Rybnik | 50.093 | 18.542 | 1 | | RY2017 | C/C (1) | I | PP525184/PP549456 |
| | | 38 | Skrajnia | 50.749 | 19.269 | 1 | | CH2016 | F/F (1) | ICE | PP549464/PP525190 |
| | | 39 | Szczejkowice | 50.098 | 18.681 | 1 | | ST2016 | C/C (1) | I | PP549459/PP525185 |
| | | **40** | **Tarnowskie Góry** | **50.443** | **18.854** | **2** | | **TG2017** | **F/- (1); C/F (1)** | **ICE, I** | **PP549458; 514 /PP525186** |
| | | 41 | Tarnów Opolski | 50.578 | 18.082 | 1 | | TO2016 | F/- (1) | ICE | PP549513 |
| | | 42 | Turawa | 50.738 | 18.069 | 1 | | TU2014 | F/- (1) | ICE | PP549515 |
| Azerbaijan | *A. colchica orientalis* | | | | | | | | | | |
| | | 43 | Zakatali | 41.600 | 46.639 | 3 | NMP-P6V 75362-1,2,3 | | | | |
| Bulgary | | | | | | | | | | | |
| | *A. fragilis* | | | | | | | | | | |
| | | 44 | Pirin | 41.609 | 23.553 | 4 | NMP-P6V 34009, 35082 | | | | |
| | *A. colchica incerta/ A. colchica* Pontic | | | | | | | | | | |
| | | 45 | Kalofer | 42.626 | 25.031 | 1 | NMP-P6V 35097 | | | | |
| | | 46 | Sozopol | 42.410 | 27.690 | 2 | NMP-P6V 34246, 33481 | | | | |
| | Grey zone | | | | | | | | | | |
| | | 47 | Vitosa | 42.538 | 23.260 | 3 | NMP-P6V 74967, 7233, 7109 | | | | |
| Czechia | | | | | | | | | | | |
| | *A. fragilis* | | | | | | | | | | |
| | | 48 | Borová Lada | 48.986 | 13.659 | 2 | NMP-P6V 34275, 32387 | | | | |
| | | 49 | Budišov | 49.267 | 15.998 | 1 | NMP-P6V 75200 | | | | |
| | | 50 | Čertovy schody | 48.631 | 14.277 | 1 | NMP-P6V 74439 | | | | |
| | | 51 | Černá v Pošumaví | 48.736 | 14.104 | 2 | NMP-P6V 35061:1-2 | | | | |
| | | 52 | Český kras | 49.938 | 14.182 | 1 | NMP-P6V 72730 | | | | |
| | | 53 | Dolní Vltavice | 48.698 | 14.076 | 1 | NMP-P6V 74408 | | | | |
| | | 54 | Dvůr Králové nad Labem | 50.429 | 15.796 | 1 | NMP-P6V 74471 | | | | |
| | | 55 | Frymburk | 48.672 | 14.178 | 1 | NMP-P6V 35096 | | | | |
| | | 56 | Horní Podluží | 50.875 | 14.549 | 2 | NMP-P6V 73650:1-2 | | | | |
| | | 57 | Horská Kvilda | 49.057 | 13.546 | 2 | MNP-P6V 35057, NMP-P6V 32640 | | | | |
| | | 58 | Hostinné | 50.533 | 15.727 | 1 | NMP-P6V 35090 | | | | |

| Country | Species | Population ID | Locality | Coordinates | | n | Museum voucher ID | ND2/RAG1 code | Genotype | ND2 haplogroups | GenBank accession number ND2/RAG1 |
|---|---|---|---|---|---|---|---|---|---|---|---|
| | | | | **N** | **E** | | | | **ND2/RAG1** | | |
| | | 59 | Hradenin | 50.030 | 15.042 | 1 | NMP-P6V 74127 | | | | |
| | | 60 | Karlštejn | 49.936 | 14.179 | 1 | NMP-P6V 71317 | | | | |
| | | 61 | Kašperské hory | 49.132 | 13.559 | 1 | NMP-P6V 35059 | | | | |
| | | 62 | Knížecí Pláně | 48.958 | 13.631 | 2 | NMP-P6V 35411 | | | | |
| | | 63 | Komořany | 49.979 | 14.418 | 2 | NMP-P6V 75663:1-2 | | | | |
| | | 64 | Královka | 50.793 | 15.163 | 1 | NMP-P6V 33728 | | | | |
| | | 65 | Lesní chalupy | 49.069 | 13.635 | 1 | NMP-P6V 75488 | | | | |
| | | 66 | Libá | 50.125 | 12.235 | 1 | NMP-P6V 35067 | | | | |
| | | 67 | Podlevín | 50.501 | 15.520 | 1 | NMP-P6V 73611 | | | | |
| | | 68 | Praha | 50.066 | 14.451 | 7 | NMP-P6V 74407, 74990, 32388, 33415, 75517, 74543, 35089/3 | | | | |
| | | 69 | Prochov | 50.464 | 15.306 | 1 | NMP-P6V 73138 | | | | |
| | | 70 | Rokytnice | 50.721 | 15.447 | 1 | NMP-P6V 73066 | | | | |
| | | 71 | Slapy | 49.822 | 14.392 | 1 | NMP-P6V 35085 | | | | |
| | | 72 | Šlechtín u Zbraslavic | 49.789 | 15.223 | 2 | NMP-P6V 73926, 75133 | | | | |
| | | 73 | Stožec | 48.880 | 13.833 | 1 | NMP-P6V 31747 | | | | |
| | | 74 | Stráž nad Ohří | 50.333 | 13.053 | 1 | NMP-P6V 72629 | | | | |
| | | 75 | Strž u Dobříše | 49.778 | 14.204 | 1 | NMP-P6V 70419 | | | | |
| | | 76 | Studenec | 49.203 | 16.050 | 1 | NMP-P6V 74875 | | | | |
| | | 77 | Vernýřov | 49.846 | 15.158 | 1 | NMP-P6V 74126 | | | | |
| | | 78 | Vlašim | 49.697 | 14.888 | 1 | NMP-P6V 71781 | | | | |
| | | 79 | Volary | 48.904 | 13.884 | 1 | NMP-P6V 31513 | | | | |
| | | 80 | Železný Brod | 50.641 | 15.252 | 1 | NMP-P6V 75606 | | | | |
| | | 81 | Želízy | 50.427 | 14.471 | 1 | MNP-6V 74410 | | | | |
| | | 82 | Zhůří | 49.081 | 13.558 | 1 | NMP-P6V 35100 | | | | |
| | | 83 | Podmoli | 48.848 | 15.943 | 1 | NMP-P6V 75111 | | | | |
| | | 84 | Potočna | 49.042 | 15.094 | 1 | NMP-P6V 74406 | | | | |
| | *A. colchica incerta* | | | | | | | | | | |
| | | 85 | Brumov-Bylnice | 49.099 | 18.006 | 1 | NMP-P6V 70642 | | | | |
| | | 86 | Grúň pod Velkou Javořinou | 48.865 | 17.684 | 2 | NMP-P6V 74469, MNP-P6V 74468 | | | | |
| | | 87 | Hostětín | 49.050 | 17.884 | 1 | MNP6V 73694 | | | | |
| | | 88 | Hukvaldy | 49.620 | 18.231 | 1 | NMP-P6V 73124 | | | | |
| | | 89 | Nové Sedlice | 49.905 | 18.006 | 2 | NMP-P6V 32368-1,2 | | | | |
| | | 90 | Slušovice | 49.244 | 17.799 | 1 | MNP-P6V 74411 | | | | |
| | | 91 | Štramberk | 49.587 | 18.120 | 15 | NMP-P6V 70591, 7415:1-3, 7419:1-2, 74153, 74103:1-2, 773238, 74132:1-3, 74121:1-2, | | | | |

**Table 1** (*continued*)

| Country | Species | Population ID | Locality | Coordinates | | n | Museum voucher ID | *ND2/RAG1* code | Genotype | *ND2* haplogroups | GenBank accession number *ND2/RAG1* |
|---|---|---|---|---|---|---|---|---|---|---|---|
| | | | | N | E | | | | *ND2/RAG1* | | |
| | Grey zone | | | | | | | | | | |
| | | 92 | Hranice | 49.549 | 17.747 | 1 | NMP-P6V 75373 | | | | |
| | | 93 | Bruntál - Mezina | 49.976 | 17.47 | 1 | NMP-P6V 74467 | | | | |
| | | 94 | Lanžhot | 48.720 | 16.964 | 1 | NMP-P6V 75343 | | | | |
| | | 95 | Lednice na Moravě | 48.795 | 16.803 | 1 | NMP-P6V 73586 | | | | |
| | | 96 | Mokrá Hora | 49.258 | 16.594 | 1 | NMP-P6V 72435 | | | | |
| | | 97 | Nejdek | 48.811 | 16.775 | 1 | NMP-P6V 35095 | | | | |
| | | 98 | Polova NPR Pálava | 48.826 | 16.673 | 1 | NMP-P6V 74426 | | | | |
| | | 99 | Otice | 49.915 | 17.870 | 1 | NMP-P6V 74466 | | | | |
| | | 100 | Littultovice | 49.899 | 17.749 | 1 | NMP-P6V 74409 | | | | |
| | | 101 | Dívčí Hrad | 50.245 | 17.636 | 1 | NMP-P6V 74759 | | | | |
| | | 102 | Králíky | 50.081 | 16.780 | 1 | NMP-P6V 33708 | | | | |
| | | 103 | Kletnice | 48.841 | 16.648 | 1 | NMP-P6V 71413 | | | | |
| **Iran** | | | | | | | | | | | |
| | *A. colchica orientalis* | | | | | | | | | | |
| | | 104 | Asalam | 37.714 | 48.959 | 1 | NMP-P6V 70057 | | | | |
| | | 105 | Motalla Sara | 38.199 | 48.870 | 1 | NMP-P6V 72678 | | | | |
| | | 106 | Nowshare | 36.650 | 51.300 | 1 | NMP-P6V 72680 | | | | |
| **Slovakia** | | | | | | | | | | | |
| | *A. colchica incerta* | | | | | | | | | | |
| | | 107 | Kriváň | 48.526 | 19.449 | 1 | NMP-P6V 35087 | | | | |
| | | 108 | Nitra | 48.346 | 18.109 | 1 | NMP-P6V 35058 | | | | |
| | Grey zone | | | | | | | | | | |
| | | 109 | Bratysława | 48.164 | 17.056 | 6 | NMP-P6V-35092:1-6 | | | | |
| **Turkey** | | | | | | | | | | | |
| | *A. colchica colchica/ A. colchica* Pontic | | | | | | | | | | |
| | | 110 | Akcakoca | 41.076 | 31.135 | 1 | NMP-P6V 70835 | | | | |
| | | 111 | Hopa | 41.379 | 41.422 | 1 | NMP-P6V 73694 | | | | |
| **Ukraine** | | | | | | | | | | | |
| | *A. colchica incerta* | | | | | | | | | | |
| | | 112 | Koneta | – | – | 1 | AC 73509 (MIZ) | | | | |
| | | 113 | Mikuliczyn | 48.406 | 24.612 | 1 | AC 74450 (MIZ) | | | | |
| | | 114 | Nadwórna | 48.618 | 24.590 | 1 | AC 74451 (MIZ) | | | | |

**Notes.**

MNHW, Museum of Natural History, Wroclaw; NMP-P6V, National Museum Praha, Czechia; MIZ, Museum and Institute of Zoology of Polish Academy of Science; F, *A. fragilis*; C, *A. colchica*; ?, unclear species assignment; Numbers in parenthesis, number of specimens with each haplotype; ICE, Illyrian-Central European; I, Carpathian lineage I; III, Carpathian lineage III; IV, Carpathian lineage IV.

Populations with detected hybrids bolded.

**Table 2** Diversity of *A. fragilis* and *A. colchica* ND2 haplotypes from Poland and summary of genetic polymorphism for both species.

| Species | Bp[1] | N[2] | η[4] | PI[5] | nH[6] | Hd ± SD[7] | Tajima D |
|---|---|---|---|---|---|---|---|
| *A. fragilis* | 729 | 0.0007 | 13 | 3 | 12 | 0.42 ± 0.077 | −2.27 |
| *A. colchica* | 732 | 0.0044 | 15 | 10 | 10 | 0.77 ± 0.061 | −0.37 |

Notes.
[1] Sequence length (Bp).
[2] Nucleotide diversity (N).
[3] Number of polymorphic sites (S).
[4] Number of mutations (η).
[5] Parsimony information sites (PI).
[6] Number of haplotypes (nH).
[7] Haplotypes diversity (Hd ± SD).
SD, Standard deviation.

bp portion exonic sequence of the nuclear recombination-activating gene 1 (*RAG1*). The protocol described by *Szabó & Vörös (2014)* was used.

PCR products were cleaned using a PCR/DNA Clean Up Purification Kit according to the manufacturer's protocols (EURX), then secondary PCR with forward and reverse primers and sequencing was done by Macrogen Inc. (Amsterdam Netherland; http://www.macrogen.com). Obtained trace files of mitochondrial and nuclear sequences were automatically assembled in CodonCodeAligner (CodonCode Corporation, http://www.codoncode.com) with limited manual correction according to the trace file.

Assembled sequences were aligned in MEGA X (*Kumar et al., 2018*) using the default settings for gap opening and extension penalties. The same program was used to estimate genetic diversity and uncorrected p-distance. All samples were translated into amino acids, which revealed no stop codons.

The dataset for *ND2* ($n = 89$) was supplemented with selected and previously published sequences from Genbank of *A. fragilis* (21), *A. cephallonica* (9), *A. colchica* (19), *A. graeca* (8), *A. veronensis* (9) and 1 *Pseudopus apodus* as the outgroup species (Table S3A) (*Gvoždík et al., 2010*; *Gvoždík et al., 2013*; *Szabó & Vörös, 2014*; *Thanou, Giakas & Karnilios, 2014*; *Jablonski et al., 2017*; *Jandzik et al., 2018*).

The dataset for *RAG1* ($n = 72$) was supplemented with *Ophisaurus attenuatus* (AY662602) as the outgroup species (*Townsend et al., 2004*) (Table S4A).

The approximate models of sequence evolution were estimated using Partitionfinder2 software (*Lanfear et al., 2017*) to find the best partition model-based results of AICc. Phylogenies were constructed using Bayesians inference (BI) performed in MrBayes (*Huelsenbeck & Ronquist, 2001*) and maximum likelihood (ML) performed in PhyML 3.3 (*Guindon et al., 2010*) with Shimodaira-Hasegawa approximate likelihood-ratio test support of branches measured (SH-aLRT) (*Anisimova et al., 2011*). Due to limited DNA evolution models in MrBayes, the following models were used according to the Partitionfinder2 results: *ND2* HKY + G for first position of the codon, GTR for second position and GTR + I for third. The same protocol was used for ML analyses. For *RAG1*, for all tree codon positions model GTR + I+G was used in Bayesians analyses. ML analyses were performed with K80 + I models, following the "Smart Model Selection" module (*Lefort, Longueville & Gascuel, 2017*).

Bayesian analyses for both markers were performed with two independent runs of Metropolis-coupled Markov chain Monte Carlo analyses. Each of the four Markov chains in temperature 0.2 ran for 700,000 generations and were sampled every 100 generations, except 25% of the first trees, which were excluded as burn-in (*Hall, 2008*)

Aligned sequences were collapsed into haplotypes in DnaSP 6 (*Rozas et al., 2017*). The same program was used to phase *RAG1* samples to gametal haplotypes (PHASE module performed with default settings), and estimated numbers of haplotypes (h), haplotype diversity (hd), number of segregating sites (S), nucleotide diversity ($\pi$), parsimony informative sites (P) and proportion between synonymous and non-synonymous mutation for mitochondrial and nuclear markers of samples from Poland. Haplotype networks were constructed using the TSC method implemented in PopART (*Leigh & Bryant, 2015*).

New nucleotide sequences have been deposited in GenBank (Table 1).

## Morphology and statistical analyses

Snout-vent length (SVL) and head proportions, the complex phenotypic characters that are known to differentiate the two studied lizards, were of special focus in this study. Along with the classical scheme of distance-morphometry, 10 characters were measured with an electronic calliper on the right side of the body, repeated three times, and a mean was used in further analyses (Tables S1A, S2A) (*Kaczmarek, 2015*; *Kaczmarek, Skawiński & Skórzewski, 2016*). As the classical technique of straight-line measurements was employed, the top-down omission of minimal and non-linear perturbations was assumed (*Humphries et al., 1981*). Well-shaped animals were exclusively taken for morphological comparisons and thus different sample sizes were used for each analysis (the numbers of specimens are written in parentheses each time). Adult slow worms of both sexes (with SVL above 120 mm, (*Sos & Herczeg, 2009*) were analysed separately to avoid the impact of clearly marked sexual dimorphism and ontogenetic development in these lizards.

Analyses of head shape were performed using transformed measurements for allometry according to the formula by *Elliott, Haskard & Koslow (1995)*. This method was used to verify the putative differences in head morphology of the two species and their hybrids. Pearson's correlation test of transformed dimensions and SVL confirmed the lack of correlations between them (Table S2B; preceded by Kolmogorov–Smirnov test).

Then, PCA was calculated on the transformed data, and a MANOVA with a Tukey post-hoc test was performed for principal component scores to compare the head shape of the two species and hybrids. MANOVA was preceded by Kolmogorov–Smirnov test and Levene's test of variance homogeneity to fulfill test's assumptions. Moreover, discriminant function analyses (DFA) were run to verify the correction of specimen classification to each group, *i.e., A. fragilis, A. colchica sensu lato* (specimens from different *A. colchica* subspecies were considered as an operational unit), or GZ. In all PCAs, the components were extracted based on a correlation matrix (*Falniowski, 2003*). The snout-vent length (SVL) between *A. fragilis* and *A. colchica* was compared with a Student's *t*-test (Levene's test of variance homogeneity was conducted before).

To describe hybrid individual morphology, identified in this study by molecular markers, in relation to the parental species, ten standard taxonomic features were evaluated including

scalation, type of prefrontal scales contact, ear opening and colouration pattern (Tables S1A and S2A). The frequencies of observed variants were analysed with chi-squared or Student's-t tests. All calculations were performed in IBM SPSS Statistics 20 (IBM Corp., Armonk, NY, USA).

## RESULTS

### Genetic differences of slow worm species and hybrids/contact zone in Poland

Two distinct diversification modes of *ND2* haplotypes were seen in slow worm species in Poland. In *A. fragilis* as many as 12 haplotypes were identified, and only one haplotype was widespread (identical to f1; *Gvoždík et al., 2010*). Some rare haplotypes were sparsely represented within the occurrence range of f1. Samples originating from eight populations carried 10 new haplotypes (Table 3). All analysed *A. fragilis* haplotypes belong to one Illyrian-Central European haplogroup (*Jablonski et al., 2016*). Nine synonymous and four nonsynonymous mutations were detected.

Within *A. colchica*, 10 *ND2* haplotypes were detected. The most frequent was c2 (*Gvoždík et al., 2010*). Samples from four populations carried five new haplotypes. In contrast to *A. fragilis*, the sequences were recognized as members of three haplogroups (I, III and IV; *Jablonski et al. (2016)* of the Carpathian lineage (Figs. 4–5; see also Figs. S1A and S1B). The uncorrected p of paired distances between groups equalled from I to IV 0.6%, from I to III 0.5% and from IV to III 0.8%. Greater haplotype diversity was noticed for *A. colchica* than *A. fragilis* (0.77 *vs.* 0.42, respectively). It is worth mentioning that specimens belonging to two *A. colchica* lineages (Carpathian I and III) co-occur in population 17 (south-eastern Poland). Eight synonymous and seven nonsynonymous mutations were detected.

Sympatric occurrence of *ND2* haplotypes of both species was noticed in two populations (nos. 29 and 40) in the central part of southern Poland (contact zone in Upper Silesia).

For *RAG1*, low genetic differentiation was noticed. Eleven haplotypes were detected, eight belonging to samples of *A. colchica* and four to *A. fragilis* based on *ND2* haplotype identification of the same specimens (Fig. S1A and S1B). Haplotype diversity for the dataset containing samples of both species were estimated to 0.61 (Hd = 0.058), and six polymorphic sites were detected (5 parsimony information sites), nucleotide diversity (Pi) were estimated to 0.00142, Tajima's D statistic: 0.404. Proportion of synonymous to nonsynonymous mutation was 1:1. Thus, in both phylogenetic analyses, samples of each taxon did not form well-separated clades/clusters, however, two main groups of sequences that correspond well with *ND2* classification were clearly shown (Fig. 6).

In both species, most of the specimens were homozygous (heterozygotes were recognized within samples of *A. colchica* from population 17 in southwestern Poland). For the two species, specific main haplotypes of wide distributions were detected: Hap_1 for *A. fragilis* and Hap_8 for *A. colchica* (Fig. 6).

Hap_1 samples form a not well-supported group (0.84/0.74) are identical to the *A. fragilis RAG1* haplotype "AfR01" from Hungary (*Szabó & Vörös, 2014*) in the analysed sequence. Within AfR01 populations a limited presence displays Hap_4 in population 11

**Table 3  List of ND2 haplotypes from Poland.** Classification of haplotypes and their population in Poland. Population ID as in Table 1. New haplotypes are bolded. N, total number of sequences in this study.

| Species | Code | Haplotype name/haplogroups | N | Population ID |
|---|---|---|---|---|
| *A. fragilis* | Hap_1 | F1 (*Gvoždík et al., 2010*) | 45 | 3, 4, 5, 6, 7, 9, 10, 11, 12, 26, 28, 29, 31, 33, 34, 35, 36, 40, 41, 42 |
| | | Illyrian-Central European | | |
| | **Hap_2** | **New** | 1 | 26 |
| | | Illyrian-Central European | | |
| | **Hap_3** | **New** | 1 | 38 |
| | | Illyrian-Central European | | |
| | Hap_4 | **DM108** (Margaryan A, GenBank Acc.no. MN122840.1) | 3 | 7 |
| | | Illyrian-Central European | | |
| | **Hap_5** | **New** | 2 | 31 |
| | | Illyrian-Central European | | |
| | **Hap_6** | **New** | 1 | 33 |
| | | Illyrian-Central European | | |
| | **Hap_7** | **New** | 1 | 33 |
| | | Illyrian-Central European | | |
| | **Hap_8** | **New** | 1 | 34 |
| | | Illyrian-Central European | | |
| | **Hap_9** | **New** | 1 | 9 |
| | | Illyrian-Central European | | |
| | **Hap_10** | **New** | 1 | 9 |
| | | Illyrian-Central European | | |
| | **Hap_11** | **New** | 1 | 13 |
| | | Illyrian-Central European | | |
| | **Hap_12** | **New** | 3 | 13 |
| | | Illyrian-Central European | | |
| *A. colchica* | Hap_1 | C4 (*Gvoždík et al., 2010*) | 1 | FJ666579.1 |
| | | Carpathian I | | |
| | Hap_2 | C1 (*Gvoždík et al., 2010*) | 12 | 15, 16, 17, 29, 30, 40 |
| | | Carpathian I | | |
| | Hap_3 | C6 (*Gvoždík et al., 2010*) | 1 | 18 |
| | | Carpathian IV | | |
| | Hap_4 | 3691 (*Jablonski et al., 2017*) | 1 | MF817483 |
| | | Carpathian III | | |
| | Hap_5 | Aro6 (*Jablonski et al., 2017*) | 5 | 19, 27 |
| | | Carpathian III | | |
| | **Hap_6** | **New** | 1 | 30 |
| | | Carpathian I | | |
| | **Hap_7** | **New** | 5 | 17, 21 |
| | | Carpathian IV | | |

**Table 3** (*continued*)

| Species | Code | Haplotype name/haplogroups | N | Population ID |
|---|---|---|---|---|
| | Hap_8 | New | 1 | 17 |
| | | Carpathian I | | |
| | Hap_9 | New | 1 | 17 |
| | | Carpathian I | | |
| | Hap_10 | New | 2 | 37, 39 |
| | | Carpathian I | | |

(new haplotype) and Hap_11 in population 18 (AfR03; *Szabó & Vörös, 2014*). Hap_8 in the analysed part is identical to AcR02 (*Szabó & Vörös, 2014*). AcR02 is widespread within all analysed *A. colchica* populations except the north-eastern part of Poland (populations 19 and 40). AcR02 was also found within samples in southern Poland (populations 18 and 17).

Most complex phylogenetic relationships of *RAG1* haplotypes were noticed for samples from population 27; all specimens represented *ND2* haplotypes, typical for *A. colchica*. A single specimen (C2017_4) represented an *A. colchica* haplotype c2 (*Gvoždík et al., 2021*) described from Finland and Lithuania (Hap_5). The other three specimens of not fully clear phylogeny represented new haplotypes (C2017_1; C2017_3: Hap_3, and C2017_2: Hap_4; Fig. 6)

The presence of two *RAG1* haplotypes common to both slow worms was detected: AfR01 (Hap_1) and Hap_7 (new haplotype). As noticed above, AfR01 was found in most specimens classified in *ND2* analyses as *A. fragilis*, and in two specimens from population 29 (1 ♀) and 40 (1 ♂), which belong to the *A. colchica* mitochondrial clade. These two specimens (TG2017_1, KK2017_1) originated from the "mitochondrial contact zone in Upper Silesia". Thus, they fit well to be considered hybrids of both species. Their hybrid origin can also be supported by values of p distance between the two specimens to *A. fragilis* (0.018%), which is 14 times smaller than to *A. colchica* (0.25%) (Fig. 6, Table S1B). Similar p distances between the group three specimens with unclear phylogeny from population 27 and *A. fragilis* is about 1.8 times smaller than that between this population and *A. colchica* (0.17% *vs.* 0.32%). Thus, these three specimens are also recognized as hybrids (C2017_1 - C2017_3), or at least display gene flow between the two species (Mazovia contact zone).

The second *RAG1* haplotype (Hap_7) shared by both species was found in four other slow worms, a single homozygous specimen from population 7 (*ND2*: *A. fragilis*) and three heterozygotes from population 17 (*ND2*: *A. colchica*). In two of them, their second sequence was nested within Hap_2 with *A. colchica* samples from population 17 (north-eastern Poland) near the main group of *A. colchica* sequences (Fig. 6). The position of the third sample (Lu2015_1_A; Hap_9) in the ML tree is unclear because this branch is not supported (0/0).

In contrast to specimens possessing Hap_1, the hybrid status of slow worms with Hap_7 is not promising. The long geographical distance between populations 7 and 17 (over 300 km air-distance), and from the potential contact zone makes their crossing unlikely.

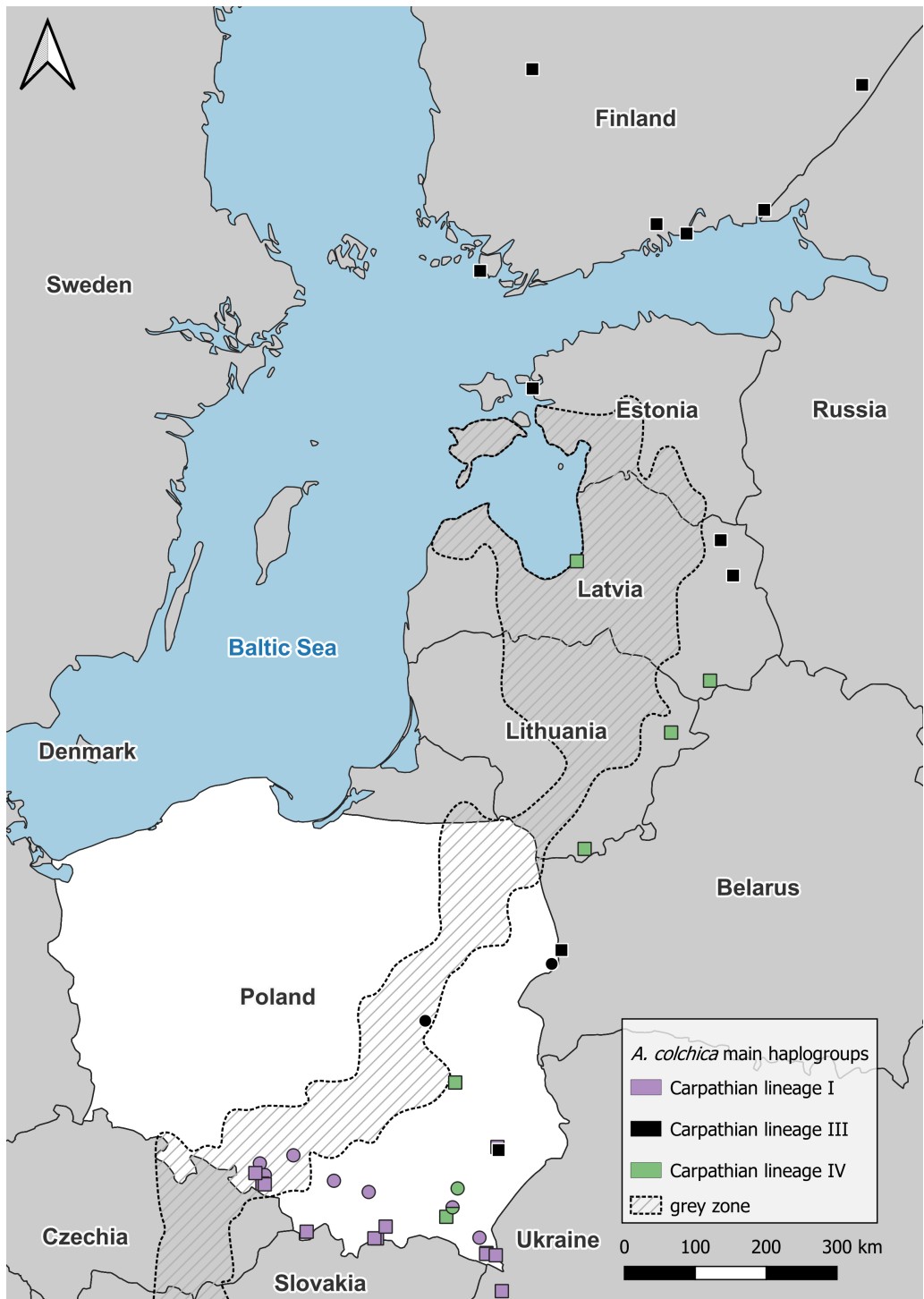

**Figure 4** *Anguis colchica ND2* **haplogroups in north-eastern Europe.** Circles, this study; squares, combined from *Jablonski et al. (2017)* and *Gvoždík et al. (2021)*.

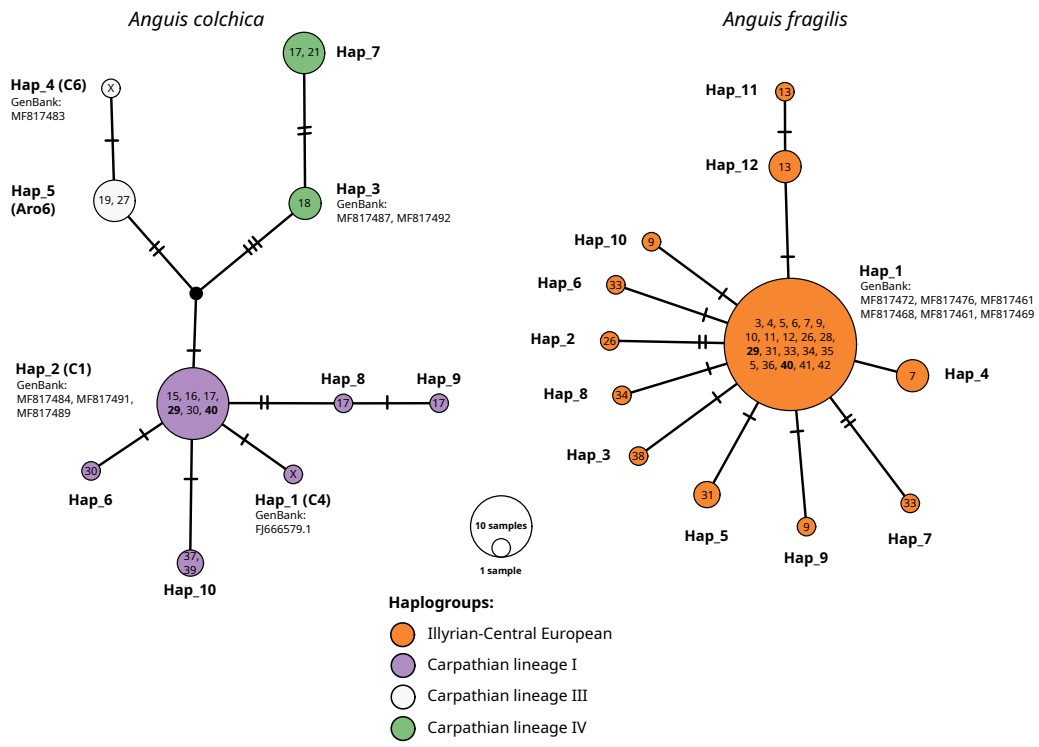

**Figure 5** **TSC haplotype networks of the *ND2* gene of slow worm samples from Poland.** The numbers in the circles refer to population ID in Table 1; X, haplotypes from GenBank, not detected in this study. Population ID with co-occurrence of both species' haplotypes bolded. Names of haplotypes are as in Table 2. Codes or names of sequences from GenBank below haplotype number. Haplogroup names after *Jablonski et al. (2016)*.

## Snouth-venth length (SVL)

In total, 116 specimens of *A. fragilis* (♂: 64; ♀: 52) and 81 specimens of *A. colchica* (♂: 41; ♀: 40) were used for comparison. The body sizes of *A. colchica* males and females (♂: M: 189 mm, SD: 27.67; ♀: M: 185 mm, SD: 37.42) are significantly larger than those of *A. fragilis* (♂: M: 168 mm, SD: 22.89; ♀: M: 162 mm, SD: 20.87) (♂: t[103] = −4.237, $p < 0.001$; ♀: t[57.383] = −3.590, $p < 0.001$; results of $t$-test with correction for variance heterogeneity).

## Head shape and interspecific classification

In separate PCA analyses performed in sex groups (*A. fragilis*: ♂: 64; ♀: 52; *A. colchica*: ♂: 41; ♀: 40; GZ: ♂: 27; ♀: 13), three main principal components with a summarized eigenvalue over 72% of the total variance were analysed (matrix of components in Table S3B). Variability patterns showed by PC-graphs (Figs. 7A and 7B) were quite similar for the two sexes, as no clear clusters were formed. Nevertheless, two overlapping groups that corresponded to species affiliation occurred, mostly along the PC2 axis, and more separate in females (Fig. 7B). GZ specimens are located within the variability of *A. fragilis*. Such a pattern was reflected in the MANOVA results (♂: F(6; 254) = 7.005; $p < 0.001$; Wilk's lambda = 0.736; ♀: F(6; 200) = 8.268; p <0.001; Wilk's lambda = 0.642) followed by Tukey's post hoc tests (Table S4B). The test results depicted the similarity of all males
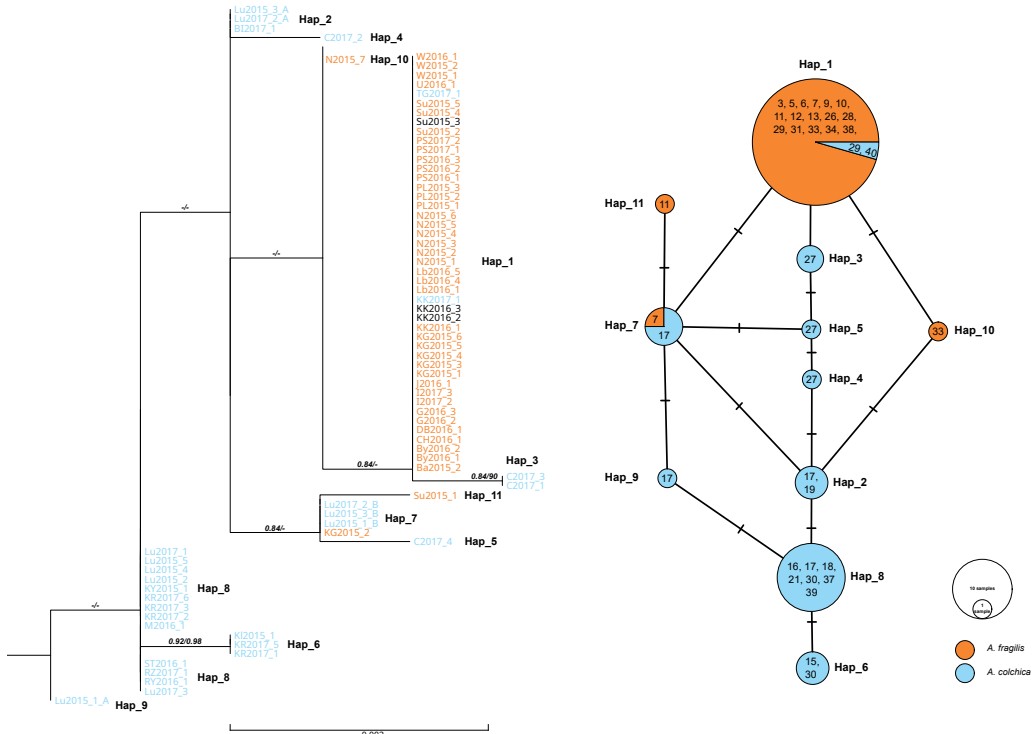

**Figure 6** **Maximum likelihood sequence tree showing phylogenetic relationships of the slow worm *RAG1* sequences (1,010 bp) from Poland (*A. colchica; A. fragilis* and putative hybrids), and corresponding haplotype network.** Classification (blue or orange colour of sequence ID) based on *ND2* analyses of the same specimens Figs. S1A and S1B, black–failed *ND2* amplification). Numbers above branches indicate the results of the Shimodaira-Hasegawa approximate likelihood ratio test of support for branches measured (SH-aLRT), followed by Bayesian posterior probability value. "-" represents no support for a branch. Numbers in circles of the *RAG1* network refer to the population ID in Table 1. Heterozygotes in *RAG1* were phased into gametic alleles, and they are given as A/B suffix at the sample code.

on PC1 and PC3, and a significant difference between the compared species on PC2 ($p < 0.001$). Moreover, a difference between *A. colchica* and GZ specimens was detected this way ($p < 0.001$). Importantly, the MANOVA assumption of variance homogeneity was not met once—for PC1 males' comparison (F(2;129) = 7.235; $p < 0.001$), so this particular result should be taken with caution. MANOVA and post hoc tests performed for females depicted significant differences on the three PCs (PC1 $p < 0.001$; PC2 $p < 0.001$; PC3 $p < 0.006$). The Tukey's test confirmed significant differences on all three PCs between *A. fragilis* and *A. colchica* females, and the distinctiveness of the GZ group on PC1 (*vs. A. colchica*, $p = 0.034$), on PC 2 (*vs. A.* colchica, $p = 0.004$; *vs. A. fragilis*, $p = 0.001$) and on PC3 (*vs. A. fragilis* $p = 0.005$) (Table S4B). It is worth noting that specimens of different *A. colchica* subspecies (*A. c. incerta*, *A. c. orientalis*, *A. c. colchica* and *A. colchica* Pontic) are distributed within variability range of *A. colchica incerta* in the PC1 *vs.* PC2 graphs (Figs. 7A, 7B).

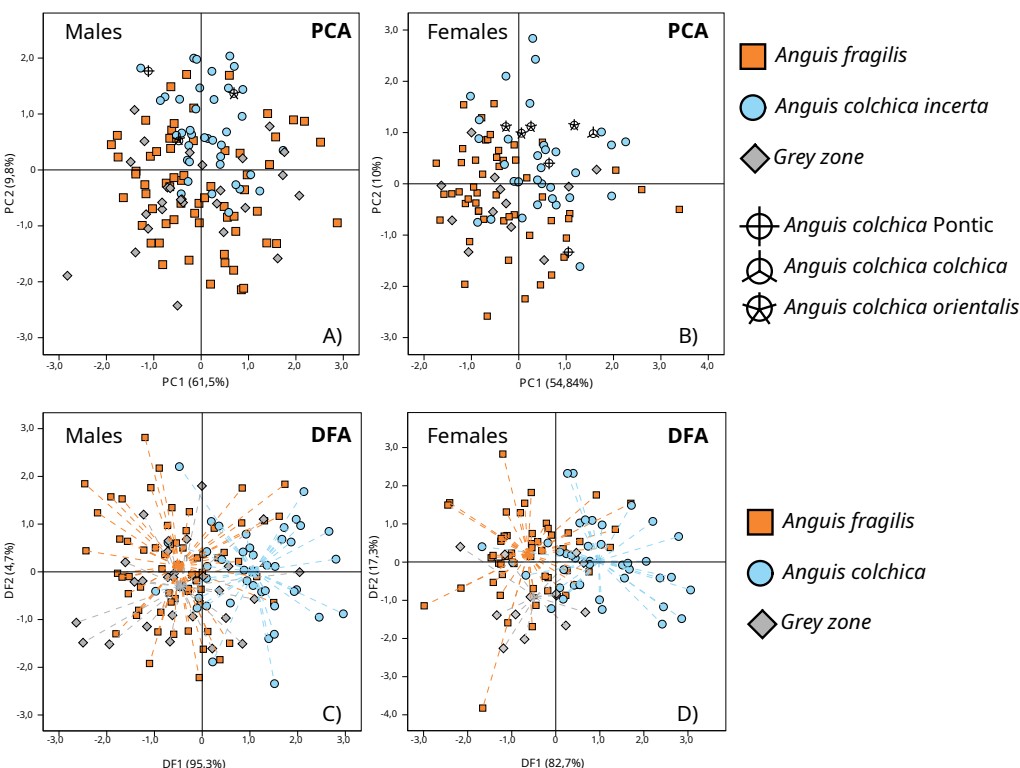

**Figure 7** **Results of principal component analyses (PCA, A and B) and discriminant function analyses (DFA, C and D) of transformed head measurements of *A. colchica*, *A. fragilis* and specimens from the grey zone (GZ).** Dashed lines (C and D) point to groups' centroids. Description of used head measurements in Table S1A. PCs matrix and eigenvalues in Table S3B. DFs eigenvalues in Table S5B, functions matrix in Table S7B.

More pronounced interspecies divergence in head morphometry was found using DFA, mainly on the first canonical function, which accounted for 95.3% and 82.7% of total variability (males and females, respectively) (eigenvalues and matrix of functions in Tables S5B and S7B) (♂: Wilk's lambda = 0.635; $df = 20$; $p < 0.001$; ♀: Wilk's lambda = 0.554; $df = 20$; $p < 0.001$) (Figs. 7C and 7D). The DFA performed for males showed that two morphological characters (HL1, and FL) influence the model most; for females there were five characters (HL1, HL2, HL3, Or-N and FL) (Table S8B). The highest correlation with canonical function 1 (CF1) was noticed for HW, HL1 and FL in males, and HL1, HL2, HL3, HW, OR_N and FL in females (Table S7B).

Cross-validated classification showed that *A. fragilis* slow worms (♂: 81.3%; ♀: 76.9%) were slightly more correctly classified than *A. colchica* (♂: 65.9%; ♀: 65%) to their respective taxa. Specimens from contact zones, as a rule, were mostly classified incorrectly (♂: 0%; ♀: 7.7%) and samples from populations 27, 29 and 40, which were identified as molecular hybrids, were assigned to *A. fragilis* in the DFA.

Table 4 **Meristic and categorial (colouration) features in genetically confirmed hybrids.** Population numbers as in Table 1; characters values and codes in Tables S1A, S2A.

| Population | Specimens ID | Sex | ND2 | RAG1 | DFA | P | EO | SRC | DP | HP | CV | CL | CT |
|---|---|---|---|---|---|---|---|---|---|---|---|---|---|
| **Celestynów (27)** | C2017-1 | ♂ | AC | ? | AF | B | 2[**] | 28[**] | 3 | 1 | 3 | 2 | 1 |
| | C2017-2 | ♂ | AC | ? | AF | C[**] | 3[**] | 26 | 1 | 1 | 3 | 2 | 1 |
| | C2017-3 | ♂ | AC | ? | AF | A[*] | 3[**] | 27[**] | 3 | 1 | 4 | 2 | 1 |
| **Kędzierzyn—Koźle (29)** | K2017-1 | ♀ | AC | AF | AF | A[*] | 2[**] | 26 | 4 | 2 | 1 | 1 | 3 |
| **Tarnowskie Góry (40)** | T2017-1 | ♂ | AC | AF | AF | B | 3[**] | 27[**] | 1 | 1 | 2 | 2 | 3 |

Notes.

AF, *A. fragilis*; AC, *A. colchica*; ?, unclear species assignment; ND2, classification of possessed haplotype; RAG1, classification of possessed haplotype; DFA, results of cross-valid classification; P, prefrontal scales position; EO, ear opening; SRC, number of scales round the body at the level of half SVL; DP, dorsal spot; HP, spots behind head; CV, abdominal colouration; CL, dorsal line; CT, dorsal/lateral border colouration.

[*]Features typical to *A. fragilis*.

[**]Features typical to *A. colchica* (*Moravec & Gvoždík, 2015*).

## Morphological relationships of hybrids to parental species

The most prominent differences between *A. fragilis* and *A. colchica* among the analysed taxonomic characters (Table S1A) were observed in (1) number of scales on the ventral part of the body (V), higher in *A. colchica*, both males and females ($M_{ACmales} = 133$ (SD $\pm$ 5.34), $M_{ACfemales} = 134$ (SD $\pm$ 5.993); $M_{AFmales} = 128$ (SD $\pm$ 5.427), $M_{AFfemales} = 128$ (SD $\pm$ 5.376)); (2) number of scale rows around the central part of the body (SRC), the same mode of differences as above ($M_{ACmales} = 28$ (SD $\pm$ 0.6), $M_{ACfemales} = 28$ (SD $\pm$ 1.01); $M_{AFmales} = 25$ (SD $\pm$ 0.715), $M_{AFfemales} = 25$ (SD $\pm$ 0.836)), (3) types of prefrontal shield contact (P), in both sexes of *A. colchica* the most frequent type was C (♂: 75.6%; ♀: 80%), and the least was type A (♂: 4.9%; ♀: 2.5%); in *A. fragilis*, types A (♂: 61.9%; ♀: 71.7%) and C (♂:11%, ♀:C = 11.3%); (4) presence of ear opening (EO), absent in *A. fragilis* males and females, and much more frequent in *A. colchica* (♂: 75.6%, ♀: 93%) (results of all performed statistical tests are summarized in Table S9B). Moreover, most of these specimens were characterized by two-side ear openings. The colour pattern is another feature that differs between the two species (Table S2A). The ventral part of *A. colchica* was clearly darker than in *A. fragilis* on the three-level scale of darkness ($CV_1 = 35\%$, $CV_2 = 47.5\%$, $CV_3 = 15\%$; $CV_1 = 15.1\%$, $CV_2 = 34\%$, $CV_3 = 50.9\%$, respectively). What's more, in females, the presence of brown spots behind the head ($HP_3$) was noticed more often in *A. fragilis* (92%) than *A. colchica* (73.7%) (Table S9B).

Considering the five hybrid specimens identified with molecular tools (populations 27, 29, and 40), the number of scales around the central body (SRC) varied between 26 and 28. All types of prefrontal shield contact were observed, and the ear opening was present in all hybrids, at least on one side of the head (two specimens). (Table 4). The colour pattern of the ventral part of the body varied somehow: two males with the light stage (CV3) (population 27), one intermediate stage (population 40), one of blue coloration ($CV_4$) (27) and a female of black coloration ($CV_1$). Brown spots behind the head were not noticed on males ($HP_1$), but they were present on females ($HP_2$).

## DISCUSSION

### The hybrid zone of slow worm species in Poland

Hybrid zone is defined as a zone where genetically distinct groups of individuals meet and mate, resulting in at least some offspring of mixed ancestry (*Harrison, 1993*). In the Polish part of the grey zone between slow worms which extends from Bulgaria in the south up to Finland in the north (*Jablonski et al., 2021*), several hybrids were detected. Hybrid individuals were detected in three populations out of 35 studied: nos. 40 (Tarnowskie-Góry; 1 ♂), 29 (Kędzierzyn-Koźle; 1 ♀) and 27 (Celestynów; 3 ♂). The first two belong to the maternal genotype of *A. colchica* (according to *ND2* haplotypes) and parental *A. fragilis* (main *RAG1* haplotype AfR01). Both are homozygotes in nuclear DNA. Likewise, the three slow worms of population 27 belong to the *A. colchica* mitochondrial clade and are homozygous in nuclear marker *RAG1*. They carry new haplotypes that are genetically closer to *A. fragilis* than *A. colchica*, especially individuals C2017_1 and C2017_3 (Table S1B, Fig. 6). It is important to note that our findings report for the first-time hybrids of maternal *ND2* haplotypes of *A. colchica* (*Šifrová, 2017*; *Benkovský et al., 2021*). Hybrid specimens between *A. fragilis* and *A. colchica* are quite rare; they were seldom reported in Hungary (*Szabó & Vörös, 2014*), Czechia and Slovakia (*Gvoždík et al., 2015*; *Šifrová, 2017*; *Benkovský et al., 2021*). Nevertheless, hybridization events can be quite common in these legless lizards because of the conservative karyotype (same number and structure of chromosomal sets) in this genus (*Altmanová et al., 2024*). Considering the microevolution of the studied lizards, the finding of homozygous *RAG1* alleles in hybrid specimens, especially those from Upper Silesia, suggests they are further level hybrids. A similar situation was reported for other lizards, *i.e., Iguana iguana* x *I. delicatissima* hybrids. They inherited nuclear alleles exclusively from *I. iguana* whereas their mitochondrial haplotypes were specific to *I. delicatissima*, so they were considered secondary hybrids rather than F1 (*Vuillaume et al., 2015*). Interestingly, the results presented by *Benkovský et al. (2021)* might deliver further examples of such individuals in the genus *Anguis* from a hybrid zone in Bratislava. All these findings imply the retained ability for successful reproduction of *A. colchica* x *A. fragilis* hybrids and lead to the conclusion that the mechanisms of reproductive isolation between those two species are insufficient or do not work efficiently. Thus, the current formal taxonomic status of the two slow worms based on the evolutionary species concept (*Benkovský et al., 2021*) seems suitable for this case. Nonetheless, the limited gene flow is observed in hybrid zones in a wide array of taxa, and it is also accepted according to the contemporary understanding of various species definitions including the biological species concept (BSC) (*Wang et al., 2019*). Transfer of some genes can be expected within *A. fragilis*—*A. colchica* species pair, because they belong to the so-colled *A. fragilis* species complex, containing four closely related European slow worm taxa: *A. fragilis, A. veronensis, A. colchica*, and *A. graeca* which began to diversify around 7 Mya and currently live in parapatry (*Gvoždík et al., 2023*). Alternatively, it may suggest the ongoing speciation of the two forms. Moreover, it is necessary to consider other explanations for the obtained results. First, a singular mutation could change a specific sequence and mislead the result because of low diversity within *RAG1*. It might explain the divergence

observed in specimens with Hap_7. Second, ancestral polymorphism in *RAG1*—the existence of a common haplotype of a progenitor inherited by the two taxa because of incomplete segregation of the new evolutionary lineages (*Avise, 2004*). This mechanism explains the presence of *A. veronensis PRLR* haplotypes from Czechia (*Šifrová, 2017*). It is worth noting that if ancestral polymorphism or single mutations are reliable explanations for the observed *RAG1* diversity, its use for *Anguis* species identification is much more limited than currently believed (*Szabó & Vörös, 2014*). On the contrary, the analysis of singular nucleotide polymorphism in populations from the Baltic region proved its use for identifying *Anguis* species (*Gvoždík et al., 2021*). Our results showed that the description of genetic differentiation between species should include a larger number of individuals from each population. More numerous study samples allowed us to recognize a higher number of *ND2* haplotypes in *A. fragilis* and in *A. colchica* from Poland (including rare sequences) than were identified in previous studies (*A. fragilis* 11 *vs.* 2, *A. colchica* 10 *vs.* 4; *Jablonski et al., 2017*). Similar results were obtained by *Harca (2021)*: 103 *ND2* haplotypes were detected (*A. colchica*—59, *A. fragilis*—44) in Czech and Slovak slow worms' populations, based on over 1,300 samples. It is worth noting that 39 haplotypes were identified from single individuals (*Harca, 2021*). Three new locations of slow worm hybrids confirm the hypothesized course of the Europe-wide *A. fragilis vs. A. colchica* hybrid zone inside grey zone (*Jablonski et al., 2021*). Populations 29 and 40 (Upper Silesia) probably constitute an extension of a hybrid zone identified in Slovakia and Czechia (*Gvoždík et al., 2015*; *Šifrová, 2017*; *Benkovský et al., 2021*; *Harca, 2021*). According to the obtained distribution data of both species' *ND2* and *RAG1* haplotypes, the Upper Silesia hybrid zone seems to be as narrow as 30–50 km. A similar size for the *Anguis* hybrid zone was found in France (*Dufresnes et al., 2023*), and even narrower (estimated to 11 km) in the Czech-Slovak contact zone (*Harca, 2021*). Putative hybrid specimens found in population 27 confirm Mazovia as the northernmost region of gene flow between the two slow worm species. Hybrid zone dynamics can be driven by more complex mechanisms *e.g.*, level and direction of gene flow for which the explanation requires further investigation, especially focusing on natural hybrid individuals.

## Morphological relationships of species and their hybrids

Although differences in size and shape of slow worms have been discussed for years (*e.g.*, *Wermuth, 1950*), the detailed morphometrical characteristics of species and their hybrids with regard to sexual dimorphism are not satisfactorily elaborated. Our data have clearly shown the larger size of *A. colchica* for both sexes. Most authors agree that *A. colchica* reaches a larger average size than *A. fragilis* (*e.g.*, *Sura, 2018*), however, the largest reported individual representing the genus *Anguis* is a male of *A. fragilis* (*Zadravec & Galub, 2018*). Recent studies have also proven a larger size of *A. colchica* than *A. fragilis*, but only for females (*Benkovský et al., 2021*). Size (and possibly shape) of the head determines the power of jaws in lizards, a trait under positive sexual selection (*Verwaijen, VanDamme & Herrel, 2002*). It was reported for *A. fragilis* that males with larger heads win combats for females (*Capula & Filippi, 1998*) and may have a stronger grasp of the female during courtship, as recorded for the common lizard *Zootoca vivipara* (*Gvoždík & Van Damme,*

*2003*). According to the aforementioned concept, *A. colchica* might monopolize all females available for mating, but there are no signs of outcompeting the second species from its area of distribution.

On the other hand, the multidimensional statistical analyses employed in this study (PCA, DFA) showed that the differences in the shapes of the heads of *A. fragilis* and *A. colchica* are not distinctive enough, and the classification algorithms are not efficient, especially for *A. colchica* (correctness of 65% at minimum). As depicted in this study, weak specificity of morphometric features of the two lizard species and their hybrids corroborates with conservative karyotypes (*Altmanová et al., 2024*) but contradicts evident genetic differentiation based on genome-wide nuclear DNA and mitogenomes (*Gvoždík et al., 2023*). Still, it must be noted that in some previous studies, such a correlation was more pronounced (*Benkovský et al., 2021*). The origin of slightly different results in independent studies might have various causes. First, the local environment can modify morphological variability, *e.g.*, *Benkovský et al. (2021)* suggested that at least different positions of prefrontal shields between species might be caused by environmental conditions or embryonic development. Second, different analytical approaches could affect the resolution of the results at genetic and phenotypic levels. Moreover, both mentioned reasons could interplay in the definite picture of phenotypic variability (*Mayr, 1969*).

As the head shape of the two slow worm species is similar, but not the same it can be expected that hybrid specimens will resemble one of the parental species more or will show intermediate head proportions. The correctness of slow worm species cross-validated classification in DFA was low; however, individuals identified as molecular hybrids were closer *to A. fragilis*. Recently published data on phenotypic differences (combining morphometric and meristic traits) between slow worm species and their hybrids also showed that specimens from the hybrid zone in Czechia and Slovakia resemble *A. fragilis* more than *A. colchica* (*Benkovský et al., 2021*). However, hybrids from the Polish part of GZ represented more *A. colchica*-like phenotypes exclusively in meristic features. Disproportions of parents' phenotypes in hybrid offspring is a common phenomenon for vertebrates, including examples from Squamata, *e.g.*, *Pituophis catenifer* and *Pantherophis vulpinus* hybrids are more like one parental taxon in head shape, but intermediate in meristic traits (*e.g.*, number of ventral scales) (*Leclere et al., 2012*). Complex genetic and epigenetic mechanisms control the expression of parental phenotype in hybrids (*Bartoš et al., 2019*), sometimes manifesting maternal dominance (*Wolf & Wade, 2009*). A closer similarity of all five molecular hybrids to the maternal parent *A. colchica* in meristic traits may result from maternal inheritance affecting the ultimate phenotype of hybrids.

It can be predicted that some future studies of the contact zones of *Anguis* species will drive to detect some new hybrid individuals. They seem to be key specimens for explaining the mechanisms that maintain the dynamics in the hybrid zones and play a crucial role in the Europe-wide distribution of slow worms.

## CONCLUSIONS

1. Clear differences between *A. fragilis* and *A. colchica* in molecular markers were proved. In phenotype, the differences were distinct in meristic features (*e.g.*, ear opening, number

of scale rows around the body) but weak in morphometrically examined head shape, especially in males.

2. Greater than the previously reported diversity in *ND2* and *RAG1* haplotypes was detected for the two species from Poland.

3. The presence of five backcross or further level hybrids was detected in Poland. This implies the reproductive activity of *A. fragilis* x *A. colchica* hybrids.

4. The five described hybrids are indistinguishable from parent species in head proportions but more resemble *A. colchica* in meristic traits.

## ACKNOWLEDGEMENTS

Thank you to RNDr. Jirí Moravec, CSc, Ph.D, for allowing access to the slow worm specimens from the collection of National Museum in Prague and Jacek Stefaniak, Ph.D, for creating the maps and improving the figures and Tomasz Skawiński, Ph.D, for helpful comments and suggestions on the earlier version of the manuscript. We appreciate the comments of two anonymous reviewers and the academic editor, which helped us to improve the manuscript.

### Funding
The authors received no funding for this work.

### Competing Interests
The authors declare there are no competing interests.

### Author Contributions
- Grzegorz Skórzewski conceived and designed the experiments, performed the experiments, analyzed the data, prepared figures and/or tables, authored or reviewed drafts of the article, and approved the final draft.
- Bartosz Borczyk analyzed the data, authored or reviewed drafts of the article, and approved the final draft.
- Stanisław Bury analyzed the data, authored or reviewed drafts of the article, and approved the final draft.
- Daniel Kulik performed the experiments, authored or reviewed drafts of the article, and approved the final draft.
- Jan Kotusz conceived and designed the experiments, performed the experiments, analyzed the data, prepared figures and/or tables, authored or reviewed drafts of the article, and approved the final draft.

### Animal Ethics
The following information was supplied relating to ethical approvals (*i.e.,* approving body and any reference numbers):

General Director of Environment Protection in Poland

## Field Study Permissions

The following information was supplied relating to field study approvals (*i.e.,* approving body and any reference numbers):

Specimens were collected in the field (Poland) thanks to the permission of the General Director of Environment Protection in Poland (No. DZP WG. 6431.02.4.2015.JRP).

## DNA Deposition

The following information was supplied regarding the deposition of DNA sequences:

The sequences are available at GenBank: PP549432–PP549520 (ND2) and PP525161–PP525231 (RAG1).

## Data Availability

The raw morphological measurements are available in the Supplementary File.

## Supplemental Information

Supplemental information for this article can be found online at http://dx.doi.org/10.7717/peerj.18563#supplemental-information.

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
