# Peer review of "Contact zone of slow worms Anguis fragilis Linnaeus, 1758 and Anguis colchica (Nordmann, 1840) in Poland"

_PeerJ, doi:10.7717/peerj.18563_

## Round 0.1 · original submission · Major Revisions

I have now received two excellent reviews that should aid you in revising and improving your manuscript. Please note the concerns of reviewer 1 about the transparency of the data that you collected. It should be very clear to the reader which specimens produced which data (morphological and molecular). Next, I want to emphasise the call by reviewer 2 to translate the ND2 sequences and ensure that your results are error free. Be realistic in your discussion about the potential for sequencing and other errors. Both reviewers have many other suggestions, including additional literature that should be included in your revision.

Reviewer 1 ·

Basic reporting

The English is generally correctly used, however some double checks or native speaker proofing are needed as some grammar mistakes like missing prepositions exist (e.g., line 241-242).
The background and references are sufficient, however, sometimes incorrectly cited like e.g. Benkowski vs. Benkovský or Jablonsky vs. Jablonski.
The manuscript is structured in a professional way, figures and tables are generally all right, just the design of Table 2 is unusual and I did not understand its structure (composed of two different sections with two headings?), and maybe the genetic polymorphism statistics for RAG1 is missing.
Raw data are shared.
Generally, the manuscript is self-contained, although aims are not clearly stated and some parts are a bit speculative (e.g. discussion about morphological variation of hybrids based on 5 individuals). On the other hand, some other potentially important subjects are missing, e.g. historical biogeography - description and interpretation of distribution of A. colchica haplogroups in the context of European distribution and historical biogeography (cf. Jablonski et al. 2017, Gvozdik et al. 2021).

Experimental design

The study is within Aims and Scope of the journal. The study is relevant and meaningful. However, it is not clearly stated how research fills an identified knowledge gap, especially in relation to the similar study by Jablonski et al. (2017).
The genetic component of this study is not based on a priori set experimental design, it is rather a descriptive study, which is acceptable, describing results based on sampling available. However, for future design, if authors would like to study hybrid zone, they should do a focused sampling across the secondary contact zone (as preliminarily delimited as “grey zone” by Jablonski et al. 2021).
The experimental design for morphological investigation is more or less good, although some issues are not clear. Most importantly, how were the presumed “hybrids” designated for morphology-only specimens? It is stated in methodology that “Specimens from previously identified contact zones were a priori classified as ‘hybrid and contact zone specimens (CZ)’ and as such subjected to morphological analyses”, but there are several morphology-only specimens marked as a parental species within the grey zone (e.g. A. fragilis in the Polish grey zone) or marked as hybrids outside the grey zone (southern Czech Rep.). This should be clarified and the “hybrids” in southern Czechia outside the grey zone should be treated as A. fragilis.
It is also not clear how the grey zone was so “smoothly” and precisely delimited in Fig. 1. Some parts are too narrow although in reality we miss data from these parts (northern Poland, Lithuania) – these parts should be wider.
One terminological issue is also necessary to make clear. “Grey zone” of Jablonski et al. (2021) is not equal to hybrid zone. It is partly equal to secondary contact zones but not only (e.g. Ireland) – it is a “grey zone” in knowledge of the distribution of Anguis species. Authors cannot confuse it with “grey zone in speciation” or hybrid zone.
Otherwise, the study performed relatively rigorous investigation with a standard technical and ethical approach.
Methods are sufficiently described, except of missing details on phasing algorithm and settings, and phasing success. Alternatively, authors should explain how potential weakly-supported gametic phases were treated. The 25% burn-in of the Bayesian analysis was based on which decision?

Validity of the findings

First, to study hybrid zone, authors need to do systematic sampling across the contact zone. The present sampling design seems to be rather random. Therefore, I suggest to CHANGE/SOFTEN THE TITLE to "Contact zone of slow worms" or "Distribution and hybridization of slow worms". Similarly, authors should soften “hybrid and contact zone” throughout the paper to “contact zone” only as they detected only 5 hybrids, and they use (correctly) the abbreviation CZ for the contact zone population.
Otherwise, the validity of the findings more or less acceptable - but it should be underlined that only 5 individuals were detected as hybrids, while designation of morphology-only "hybrids" might be imprecise and these "hybrids" could also be composed of "pure" parental species.
I would also strongly recommend to compare morphologically the different colchica subspecies separately, including the candidate “Pontic” subspecies from Sozopol (BG) and Akcakoca (TR), especially considering that A. colchica incerta and A. colchica colchica (+ orientalis + “Pontic”?) might represent two distinct species (see Gvozdik et al. 2023). So, authors should be careful with interpretation of morphological data of specimens from Turkey, Iran, Azerbaijan and part of Bulgaria. The comparison of the species as currently understood (fragilis vs. colchica) is acceptable, but should be also accompanied by a comparison of fragilis vs. individual ssp. of colchica or at least colchica incerta (and avoid the other subspecies).
Next, authors should also consider the issue (incorrectly discussed throughout the paper) that F1 hybrids are not possible to find in the case of two species forming a hybrid zone. The population in the hybrid zone is a hybrid population with a steep but gradual cline of genotype composition of the parental species and F1 hybrids cannot be present in the hybrid zone. This should be considered also in conclusions. The last point of conclusions should be softened too, as it is based on 5 individuals only.
Furthermore, in Table 1, it is confusing that genotypes are not always listed for all individuals. For example, the population 12 was sampled by 7 individuals but only 2 genotypes are given, moreover one as missing for RAG1. Either all genotypes should be listed or at least the number of individuals bearing the particular genotype should be given. For the hybrids, I prefer to list all genotypes for all the five individuals, see populations 41, 42. What is “H” in RAG1 genotype in pop. 41 (also Table 3), “hybrid genotype”? But all 5 hybrids were said to carry homozygous RAG1 genotype…!?
Next, authors use Szabo & Vörös’s RAG1 haplotype codes, however, in reality, they confused these codes with ND2 codes and did not use the correct Szabo & Vörös’s RAG1 haplotype codes. This should be corrected. Authors should also compare whether the “new” haplotypes were not previously found by Gvozdik et al. (2021).

Additional comments

I strongly suggest to combine the authors’ new data with previously published data by Jablonski et al. (2017), including graphically in Fig. 1. It would be also beneficial for readers and historical biogeographic interpretations) within the range-wide scale (which are missing from the present state of the manuscript) to include graphically the information on the distribution of A. colchica haplogroups (see Jablonski et al. 2017, Gvozdik et al. 2021).
Some potentially important publications are missing in introduction, e.g. Jablonski et al. (2016), Thanou et al. (2014). If authors want to cite master theses (e.g. Sifrova 2017), there is one more potentially important master thesis – Harca (2021), accessible at https://is.muni.cz/th/yeu1r/DP_Harca_FINAL_sprilohou.pdf?info. It should be noted in the reference list that the theses are in Czech. Furthermore, as authors included populations from the Czech Republic, they might want to study and cite the book by Moravec (2015, ed.) on the reptile fauna of Czechia (Fauna of the Czech Republic, Reptilia), chapters are cited individually, see Jablonski et al. (2021).
Authors should remove from discussion “the current formal taxonomic status of the two slow worms can be put into question” as there is no supporting evidence for such a statement. Moreover, there should be “slow-worm species”, if this sentence is used.
Authors might want to rephrase the last paragraph of discussion. To speak about “discovery of new hybrids” sounds strange in a species pair forming a hybrid zone.

Minor issues:
Be careful to distinguish between specimen and individual. If you speak about museum voucher or tissue sample, DNA, then it is specimen, but otherwise use individual.
References: “Javurkova” in Benkovsky et al. should be listed as “Gvozdikova Javurkova”
NMP6V is currently used as NMP-P6V, please, change this museum code.
GenBank numbers are better to list full, not only the last two/three digits.
For the coordinates of the two Iranian localities with missing geo-info, see Gvozdik et al. (2010).
Ukrainian “Koneta” might be “Konela” or “Koneva”.
Table 3, legend: add (a) reference(s) for “features typical”; authors might want to read the book (relevant chapters) edited by Moravec (2015).
Fig. 1 could have an alternative version in supplementary material where all populations will be numbered.
Fig. 2: Authors might consider to use the same colour coding as in Jablonski et al. (2017) that the data are more easily comparable.
Fig. 3: Hap_2 is given twice (probably instead of Hap_6), change Anguis sp. to Anguis spp., and further I am afraid that haplotype codes will not be readable, authors should think about an alternative (maybe homozygotes might be shown only as one haplotype). Why authors use black for fragilis if red is used elsewhere? This is confusing, change red and black.
Fig. 4: Similarly, why yellow for colchica (green elsewhere) and blue for CZ (could be “neutral” like grey or white)? Or authors might want to follow the colour coding as in Benkovsky et al. (2021) to make morphological comparisons easier.

Change:
“morphological markers” to “morphological characters”
“dimorphic traits” to “sexually dimorphic traits”
“squamation” to “scalation”
“influence” to “affect”, at least on line 412
line 311 “samples” to “populations”
line 374 “designated” to “identified”
line 423 “Squamate” to “Squamata” or “squamates”
line 424 “taxa” to “taxon”
line 428 “mother species” to “maternal parent” or similar
“Bratyslawa” to “Bratislava”
“approximate ratio” to “approximate likelihood-ratio”

Reviewer 2 ·

Basic reporting

In this MS the authors analyzed genetic diversity and morphology of two species of legless lizards Anguis fragilis and A. colchica and their putative hybrids from the area of their secondary contact in Poland. The study is a follow-up to a similar study previously conducted by another team in the nearby region of Czechia and Slovakia. The authors employed a compatible design of the study, which allowed them for broader comparisons. Their results are largely corroborative, with a few exceptions where new patterns were observed, such as higher similarity of the putative hybrids to A. colchica instead of A. fragilis in some meristic characters. Surprisingly, the authors also recorded significantly higher mtDNA diversity than previously recorded for either species.

The MS is straightforward, clearly written, although the language would benefit from a polishing by a more experienced English writer. The analyses are adequate to the collected data and interpretations are usually based on the original results.

My main concern is with a minor theme in the MS, however providing the most distinctive new result. It is the high ND2 haplotype diversity. The significantly higher ND2 haplotype numbers in both species than previously reported are rather surprising. In particular, because 1) the numbers of the analyzed individuals were relatively low, 2) their majority comes from the areas very close to the region in Czechia and Slovakia with much lower diversity, and 3) the reported diversity increase is not mirrored in nuDNA sequences. In most cases the new haplotypes were identified from single individuals. I understand that questioning the quality of the raw data is walking on a thin ice, nevertheless my suspicion that PCR and/or sequencing errors could explain at least a portion of such "diversity increase" is too strong to shy away from raising my concerns about it. So, how did the authors control for this? Have the observed SNP frequencies deviated from the expected error rate of the used PCR polymerase and/or sequencing pipeline? What is the proportion of non/synonymous mutations? Have the sequences been translated to check whether the putative mutations make sense? Did you check the histograms and excluded low quality sequences? Did you repeat PCRs and/or sequencing of the samples containing the new haplotypes or have this only been done once?

My second concern is with the proper use of statistical tests. The authors do not mention whether the collected morphometric, meristic, and categorical datasets conform with the particular test assumptions and/or whether any transformations beside the control for allometry were deployed before running the analyses. Specifically, the information about the patterns of variance and data distribution should be clear from the test descriptions.

A few minor concerns:
34 - inefficient mechanisms, rephrase
48 - veronensis, not veronesis
46-51 - N Italy and S France are treated as one region here, however, Czechia, Slovakia and Hungary are three separate ones, even though the zone of sympatry and hybridization is clearly just one here. Maybe instead of talking about vaguely definied regions, list the species pairs that do co-occur/hybridize?
76 - Benkovský, not Benkowski
79 - Please double check the methods in Benkovský et al.; I don't believe the study genotyped all morphologically analyzed individuals. Rephrase.
109 - I assume you mean dissection here? Not histological sectioning, correct? Please rephrase.
164 - Please add the GB Acc. Nos. here or a reference to where in the paper they can be found.
185-186 - SVL is not a measure of the entire body size, please rephrase.
186 - Before running the t-test, where the datasets tested for homoskedasticity and/or normal distributions/s?
204 - uncorrected instead of uncorrelated
258 - Please bs specific what measure in "body size".
259 - than those instead of than that if phrased in plural
262 - I suggest removing the word "shape" as this was not exactly analyzed, rather it was separate dimensions of the head.
263 - Isn't the word "sex" more suitable than "gender" in this context?
267 - A weak tendency to form? What does this mean? Please be specific and refrain from using dynamic words to describe static observations.
283 - HL1, HL1? I assume that's a type, please fix.
296-317 - in the entire chapter results of the statistical tests are missing, though they are provided in the previous chapter. If it makes too complicated to list them directly in the text, please put references to where result of each test could be found.
298 and further - are these means? If so, please always list mean with SD when doing comparisons
302 - the frequencies of P types do not add up in Ac females - C+A make 105%, was it maybe supposed to be 2.5 in C?
321 - "Hybrid zone" instead of "A hybrid zone"
325 - Well, the localities are not the hybrid zones, they can belong or lie within a hybrid zones; please rephrase.
328 - RAG1 genotype could be both paternal and maternal, you cannot conclusively claim it is a paternal genotype even in a homozygote.
358 and further - Isn't it more suitable to use the term single/spot mutation (polymorphism) than singular mutation (polymorphism)?
371 - Jablonski, not Jablonsky
415-416 - Please rephrase, this sentence does not make sense and I'm not sure what tha authors mean here. Similar does not mean same. With transgressive, do you mean overlapping? Why cannot it be expected? I would argue that it logically could.
417 - DFA unsuccessful; please be specific, did not show results at a certain level of significance or the results did not confirm the null hypothesis?
423 - squamates or Squamata not Squamate
424 - taxon not taxa
421-429 - I think the authors might consider discussing alternative scenarios here, e.g. about differently directed gene flow that would explain the observed pattern different from that in other parts of the hybrid zone.
430-431 - If I understand correctly, the grey zones in Jablonski et al. 2021 are not necessarily hybrid zones but rather secondary contact zones.
439 - dimensions, not shape, see above
440 - Greater than previously reported?
443 - This is not precise. It could be the present or historical backcross hybridization, not necessarily a recent one; please rephrase.
Fig. 3 - the labels in the tree and to a lesser extent in the network are almost unreadable. Please increase the font size.
Fig. 4 - ditto

Experimental design

-

Validity of the findings

-

Additional comments

-

---

## Round 0.2 · Minor Revisions

Thanks for your revision. Please note the reviewers' important request not to confuse the stipulated area with a taxonomic trinomial. Please ensure that the supplementary information is updated.

Reviewer 1 ·

Basic reporting

Thanks to the authors for this much improved manuscript. I am happy with their responses and the revision.

I only have minor comments regarding rather technical issues:

Minor but VERY IMPORTANT: The authors must refer to the Pontic population of A. colchica as “A. colchica Pontic” (Pontic with capital P and NOT in italics!), or in a similar way, and not in the way which looks like a scientific trinomen, i.e. subspecies name, which is the present case and is ERRONEOUS. It must be corrected EVERYWHERE, including in supplementary materials! Otherwise, the authors introduce a nomen nudum, taxonomically invalid name but bringing confusion.

Legend to Fig. 3: I suppose the reference should be Jablonski et al. (2017) and not “2021”.

Legend to Fig. 4: Again, I suppose the reference should be Jablonski et al. (2017) and not “2021”, and maybe it would be better “in north-eastern Europe” instead of “Baltic area”. The authors could also consider keeping only names of countries, which are discussed in the text, and maybe the whole figure could be better focused/zoomed on the grey zone and nearby areas without showing e.g. Germany, Norway. In addition, symbols are relatively small and the white ones not much highlighting (easy to overlook).

Fig. 7: “Anguis colchica sensu incerta” should be “Anguis colchica incerta” and “Anguis colchica sensu lato” just “Anguis colchica” or alternatively “Anguis colchica sspp.” (plural for subspecies). Make sure that the symbols for A. c. colchica, orientalis and Pontic are readable. Now, in the pdf version for review, it is not distinguishable due to lower resolution.

I am not certain whether I have overlooked it, but I am missing legends to the supplementary files.

Further, it seems that for example Fig. S1 is even not referenced in the text.

Experimental design

-

Validity of the findings

-

Additional comments

-

---

## Round 0.3 · accepted · Accept

Thank you for your revisions.
Please note that there are still many grammatical errors. Please use something like Grammarly.com to check for errors throughout the text and legends - and correct them.

L84: genetic identification of considerable number of specimens = genetic identification of a considerable number of specimens
L454: grammatical and bracket error.
L460: please remove numbering and make into a complete paragraph with full sentences. Currently, 1 & 2 are not complete sentences.
L466: was = were